# Effect of Zwitterionic Additives on Solvation and Transport of Sodium and Potassium Cations in (Ethylene Oxide)_10_: A Molecular Dynamics Simulation Study

**DOI:** 10.3390/nano14020219

**Published:** 2024-01-19

**Authors:** Manh Tien Nguyen, Yuhua Duan, Qing Shao

**Affiliations:** 1Chemical and Materials Engineering Department, University of Kentucky, Lexington, KY 40506, USA; 2National Energy Technology Laboratory, United States Department of Energy, Pittsburgh, PA 15236, USA; yuhua.duan@netl.doe.gov

**Keywords:** zwitterion, sodium-ion batteries, potassium-ion batteries, ionic solvation, molecular dynamics

## Abstract

Sodium- (Na^+^) and potassium- (K^+^) ion batteries are cost-effective alternatives to lithium-ion (Li^+^) batteries due to the abundant sodium and potassium resources. Solid polymer electrolytes (SPEs) are essential for safer and more efficient Na^+^ and K^+^ batteries because they often exhibit low ionic conductivity at room temperature. While zwitterionic (ZW) materials enhance Li^+^ battery conductivity, their potential for Na^+^ and K^+^ transport in batteries remains unexplored. In this study, we investigated the effect of three ZW molecules (ChoPO4, i.e., 2-methacryloyloxyethyl phosphorylcholine, ImSO3, i.e., sulfobetaine ethylimidazole, and ImCO2, i.e., carboxybetaine ethylimidazole) on the dissociation of Na^+^ and K^+^ coordination with ethylene oxide (EO) chains in EO-based electrolytes through molecular dynamics simulations. Our results showed that ChoPO4 possessed the highest cation–EO_10_ dissociation ability, while ImSO3 exhibited the lowest. Such dissociation ability correlated with the cation–ZW molecule coordination strength: ChoPO4 and ImSO3 showed the strongest and the weakest coordination with cations. However, the cation–ZW molecule coordination could slow the cationic diffusion. The competition of these effects resulted in accelerating or decelerating cationic diffusion. Our simulated results showed that ImCO2 enhanced Na^+^ diffusion by 20%, while ChoPO4 and ImSO3 led to a 10% reduction. For K^+^, ChoPO4 reduced its diffusion by 40%, while ImCO2 and ImSO3 caused a similar decrease of 15%. These findings suggest that the ZW structure and the cationic size play an important role in the ionic dissociation effect of ZW materials.

## 1. Introduction

As promising alternatives to lithium-ion (Li^+^) batteries, sodium- and potassium-ion (Na^+^ and K^+^) batteries have garnered researchers’ attention because of the abundance and affordability of sodium and potassium resources [1,2]. However, the development of Na^+^/K^+^ battery technology is still in its early stages and faces several challenges. One of the primary obstacles is the development of safe and efficient electrolytes [1]. For long-term battery performance, the electrolyte must be electrochemically stable across a wide potential range during charge–discharge cycles [2]. Additionally, it should be compatible with both the anode and the cathode materials. Some electrolytes may react with the electrode materials, leading to side reactions and degradation on the electrode surface [3,4,5]. Another critical challenge is the formation of dendrites during battery cycling with liquid-phase electrolytes [2,6,7]. Dendrites can breach the separator, causing short circuits and compromising safety, as well as reducing battery lifespan [5]. The identification of electrolytes that can suppress or mitigate dendrite formation is crucial for the realization of Na^+^ and K^+^ batteries.

Solid polymer electrolytes (SPEs) have the potential to enhance the performance and safety of Na^+^ and K^+^ ion batteries. In comparison to liquid electrolytes, SPEs typically exhibit a broader electrochemical stability window and may inhibit dendrite formation due to their higher mechanical properties [2]. However, SPEs, particularly those based on poly(ethylene oxide) (PEO), generally possess low room-temperature ionic conductivity [8,9,10,11]. As the ionic transport efficiency affects the overall battery performance, enhancing ionic conductivity in SPEs is imperative for improving the efficiency, safety, and overall feasibility of Na^+^ and K^+^ ion batteries.

Zwitterionic (ZW) materials have been applied in SPEs for Li^+^ batteries with enhanced conductivity. ZW molecules have unique structures and properties because they have both positive and negative charges within the same molecule [12,13,14]. Such characteristic enables ZW molecules to interact with both Li^+^ and anions, leading to enhanced solvation of Li^+^ [15,16,17], which, in turn, promotes the dissociation of Li^+^ from the polymer backbone and facilitates ionic mobility. ZW materials have been reported with higher ionic conductivity compared to conventional SPEs [18,19,20,21,22]. For example, a ZW polymer scaffold—poly(carboxybetaine methacrylate)—doubled Li^+^ ionic conductivity in LiTFSI/EO_5_ electrolytes [21]. In another study, Segalman and co-workers developed ZW SPEs that exhibit high Li^+^ conductivity (1.6 mS/cm) and Li^+^ transference number (t_Li+_ ≈ 0.6–0.8) [22].

Molecular dynamics (MD) simulations have provided valuable insights into ZW molecules’ ability to regulate Li^+^ solvation and transport. In our prior study [23], we found that ZW molecules influenced Li^+^ diffusion in EO-based electrolytes via two opposing mechanisms: (1) by releasing Li^+^ from the wrapping effect of EO chains and accelerating Li^+^ transport; (2) by trapping Li^+^ in the strong Li^+^–ZW molecule coordination and decelerating Li^+^ transport. The interplay of these two effects correlated with the EO chain length: the accelerating effect was enhanced with the increasing EO chain length. Additionally, the dissociation effect of ZW molecules was contingent on their structural composition, especially their anionic group. Three distinct ZW molecules dissociated Li^+^ from Li^+^–EO_10_ coordination with different efficiency, in the following order: ChoPO4 > ImCO2 > ImSO3 (ChoPO4: 2-methacryloyloxyethyl phosphorylcholine, ImSO3: sulfobetaine ethylimidazole, and ImCO2: carboxybetaine ethylimidazole) [24]. Among these three ZW structures, ChoPO4, characterized by a PO_4_ anionic group, exhibited the highest electronegativity, while ImCO2, with a CO_2_ anionic group, possessed the highest charge density (ImSO3 featured a SO_3_ anionic group). Besides, computational simulation results suggest that ZW additives benefit Li^+^ transport in high-Li^+^-concentration environments [24]. All three ZW molecules reduced Li^+^ diffusion at a low Li^+^ concentration (Li^+^/O(EO_10_) molar ratio – r = 1:18), while at a higher Li^+^ concentration (r = 1:6), only ImSO3 slowed down Li^+^ diffusion [24].

Although the effect of ZW molecules in Li^+^ electrolytes has been investigated, their potential in regulating Na^+^ and K^+^ solvation and transport has yet to be understood. Particularly, it remains unclear whether ZW compounds can effectively dissociate Na^+^ and K^+^ from EO chains and impact their ionic transport with the same mechanism as that observed for Li^+^. Further study is needed to comprehensively assess the capability of ZW molecules to enhance the performance of Na^+^ and K^+^ electrolytes.

This work investigated the dissociation effect of ZW molecules on Na^+^/K^+^ for their possible application to regulate ionic solvation and transport. This work used NaTFSI/EO_10_ and KTFSI/EO_10_ as model systems ([TFSI]^−^, bis(trifluoromethylsulfonyl)-imide)). Three ZW molecules (ChoPO4, ImCO2, and ImSO3) were applied in our simulation systems. To explore the effects of ZW molecules on Na^+^/K^+^ transport, we performed MD simulations on two systems without ZW molecules, i.e., NaTFSI/EO_10_ and KTFSI/EO_10_ at Na^+^/O(EO_10_) and K^+^/O(EO_10_) molar ratios (r) = 1:6; and six systems with ZW molecules, i.e., NaTFSI/EO_10_/ZW molecule and KTFSI/EO_10_/ZW molecule systems at Na^+^/ and K^+^/ZW molecule molar ratios = 5:1.

## 2. Simulation Methods

### 2.1. Molecular Model

This study utilized the all-atom model to describe all ions and molecules. Figure 1 shows the molecular structures of three ZW compounds, [TFSI]^−^ ions, and EO_10_ molecules. The bonded potential includes bond, angle, and dihedral angle potentials. The nonbonded potential is the sum of the Lennard–Jones 12-6 potential and the Coulomb potential (Equation (1)):(1)$$E=\sum_{i}\sum_{j<i}\left\{\frac{1}{4\pi \varepsilon_{0}}\frac{q_{i}q_{j}e^{2}}{r_{ij}}+4\varepsilon_{ij}\left[{\left(\frac{\sigma_{ij}}{r_{ij}}\right)}^{12}-{\left(\frac{\sigma_{ij}}{r_{iJ}}\right)}^{6}\right]\right\}$$
where *r_ij_* is the distance between the atoms *i* and *j*, *q_i_* and *q_j_* are the partial charges of the atoms *i* and *j*, *ε*_0_ is the free space permittivity, and *ε_ij_* and *σ_ij_* are energetic and geometric parameters. The bonded and nonbonded interactions are described by the OPLSAA/MM force field [25]. The force field parameters were generated by the Ligpargen web server [26]. Appendix A lists the nonbonded force field parameters used in this work.

The initial simulation systems were generated by randomly positioning ions and molecules within a cubic box using Packmol [27]. Table 1 lists the details of the eight simulation systems. Figure 2 shows the configuration of a simulation box containing 200 Na^+^ (or K^+^) ions, 200 [TFSI]^−^ ions, 40 ImSO3 molecules, and 120 EO_10_ molecules.

### 2.2. Simulation Details

The simulation process of each system consisted of six steps. Initially, an energy minimization was conducted to prevent atoms from becoming too close. Second, a 100 ns isothermal–isobaric (NPT) MD simulation was carried out at 500 K and 100 kPa to thoroughly relax the system. Subsequently, a third NPT simulation of 200 ns was conducted at 353 K and 100 kPa, followed by a fourth canonical (NVT) simulation of 1000 ns at 353 K. In the fifth step, a 200 ns NVT simulation was performed to equilibrate the system at 600 K. Finally, a canonical (NVT) simulation of 1000 ns was conducted at 600 K for data collection at every 5 ps. The purpose of steps 3 to 6 was to ensure that the systems closely approximated the desired density at 353 K. Moreover, the high temperature of 600 K accelerated the mobility of ions and molecules within the systems, enabling a more effective observation of the systems’ structural and dynamic properties within a reasonable simulation time frame of 1000 ns. To regulate the temperature and pressure in the second, third, and fifth steps, the Berendsen method [29] was employed due to its ability to bring the system to the specified temperature and pressure quickly. In the fourth and sixth steps, the system’s temperature was controlled using the velocity-rescaling method [30]. Short-range potential energies were calculated using a 1.0 nm cut-off, while the long-range electrostatic interaction energy was determined using the particle mesh Ewald sum [31]. During the fourth, fifth, and sixth steps, all bonds involving hydrogen atoms were constrained using the LINCS algorithm [32]. The simulations were repeated three times for each system, employing different initial configurations. Gromacs 2021 software [33] was used to conduct the energy minimization and MD simulations.

### 2.3. Analysis Methodology

Radial distribution functions (RDFs) or pair correlation functions g(r) are used to describe how the density varies as a function of the distance from a reference particle. To investigate the impact of ZW molecules on the coordination preference of Na^+^ and K^+^, we analyzed the RDFs of cation–O([TFSI]^−^), cation–O(EO_10_), and cation–O(ZW) in the eight electrolyte systems listed in Table 1. O([TFSI]^−^) denotes all four O atoms of the [TFSI]^−^ molecule. O(EO_10_) denotes all ten O atoms of the EO_10_ molecule. O(ZW) denotes four O atoms (O3, O4, O5, and O6) of the –PO_4_ group in the ChoPO4 molecule, two O atoms (O1 and O2) of the –CO_2_ group in the ImCO2 molecule, and three O atoms (O1, O2, and O3) of the –SO_3_ group in the ImSO3 molecule (Appendix A).

The coordination numbers of cation–O(EO10) (N_cation-O(EO10)_), cation–O([TFSI]^−^) (N_cation-O([TFSI]−_), and cation–O(ZW) (N_cation-O(ZW)_) were derived through numerical integration of the corresponding RDFs within a threshold, which was determined based on the first valley of the RDFs (Na^+^–O(EO10): 0.40 nm, Na^+^–O([TFSI]^−^): 0.35 nm, Na^+^–O(ZW): 0.30 nm, K^+^–O(EO10): 0.40 nm, K^+^–O([TFSI]^−^): 0.40 nm, and K^+^–O(ZW): 0.35 nm).

We computed the percentage of cations that were not in coordination with O(EO_10_) in the eight systems. The cations not coordinated with EO_10_ were those having no O atoms from EO10 in their first coordination shell, which was defined by the first valley in the RDF figures (Na^+^–O(EO10): 0.40 nm, Na^+^–O([TFSI]^−^): 0.35 nm, Na^+^–O(ZW): 0.30 nm, K^+^–O(EO10): 0.40 nm, K^+^–O([TFSI]^−^): 0.40 nm, and K^+^–O(ZW): 0.35 nm).

We assessed the stability of the cation–O(EO_10_), cation–O([TFSI]^−^), and cation–O(ZW) coordination through their respective lifetimes (τ_cation-O(EO10)_, τ_cation-O([TFSI]−)_, and τ_cation-O(ZW)_, respectively). A higher lifetime τ indicated a more stable coordination. The lifetime τ is computed by integrating the coordination residence curves (*C*(*t*)). The *C*(*t*) curves are constructed according to
(2)$$C\left(t\right)=\frac{1}{N_{0}}\sum_{j=0}^{N_{0}}\frac{\langle P_{j}\left(0\right)P_{j}\left(t\right)\rangle }{{\langle P_{j}\left(0\right)\rangle }^{2}}$$
where *N*_0_ is the initial total number of a specific coordination at time *t* = 0, with $P_{j}\left(0\right)=1$. $P_{j}\left(t\right)$ is set to 1 if the coordination *j* persists and becomes 0 if the coordination *j* breaks at time *t*. The corresponding *C*(*t*) curves are shown in Appendix A.

The diffusion coefficients were computed using the Einstein method, which involves analyzing mean-square displacement (MSD) curves
(3)$$D=\underset{t\to \infty }{\lim}\frac{1}{6t}\langle {\left|r\left(t\right)-r(0)\right|}^{2}\rangle$$
where *<|r(t) − r(0)|^2^>* denotes the mean square displacement of an atom or molecule species, with *r(0)* and *r(t)* being the positions at time *0* and *t*, respectively. The MSD curves are shown in Appendix A. Appendix A list detailed information on the diffusion coefficients of cations, [TFSI]^−^, and ZW molecules.

The short-scale diffusion of cations versus cation-O coordination is quantified by measuring the distance traveled by cations that maintain consistent coordination with a specific number of oxygen atoms from ZW molecules, EO_10_, or [TFSI]^−^ within a 1-ns time frame. In addition to the 1 ns interval, we extended the analysis to a 2 ns duration to further support our conclusion regarding the influence of varying cationic coordination on the short-scale motion of cations (Appendix A).

We analyzed EO_10_ conformation using the radius of gyration (*R*_g_), a measure of the average coil size of EO_10_ chains. *R*_g_ is calculated as follows:(4)$$R_{g}=\sqrt{\frac{\sum_{i}m_{i}r_{i}^{2}}{\sum_{i}m_{i}}}$$
where *m_i_* is the mass of atom *i*, and *r_i_* is the position of atom *i* relative to the center-of-mass of the molecule.

## 3. Results and Discussion

### 3.1. Radial Distribution Functions (RDFs)

Figure 3 shows the cation–O([TFSI]^−^), cation–O(EO_10_), and cation–O(ZW) RDFs in Na^+^ and K^+^ electrolytes. The RDFs of Na^+^–O([TFSI]^−^), Na^+^–O(EO_10_), and Na^+^–O(ZW) showed a high peak located around 0.24 nm, consistent with the literature [34,35,36,37,38,39,40]. Meanwhile, the K^+^–O([TFSI]^−^) and K^+^–O(EO_10_) RDFs showed high peaks at a further distance of around 0.27 nm. Previous studies [38,41,42,43,44,45] also reported that the K⁺–O RDF peaks were at a greater distance at around 0.28 nm, indicating weaker K⁺–O interactions compared to Na^+^–O interactions. The high RDF peaks suggested a strong coordination between Na^+^ and K^+^ ions with [TFSI]^−^, EO_10_, and ZW molecules.

Among these three ZW structures, ChoPO4 demonstrated the most pronounced effect in reducing the preference of Na^+^ for coordinating with [TFSI]^−^ and EO_10_. Figure 3a,b shows that the RDF peak heights of Na^+^–O([TFSI]^−^) decreased from 10 to 7 with ChoPO4 addition, whereas were around 8 and 9 with ImCO2 and ImSO3, respectively. Likewise, the Na^+^–O(EO_10_) RDF peak height decreased from around 14 to 12 with ChoPO4 and to about 13 with ImCO2 and ImSO3. A similar phenomenon was also observed in the case of Li^+^ in our prior study [24], which showed that ChoPO4 presence resulted in a 30% decrease in the RDF peak heights of Li^+^–O(EO_10_) and Li^+^–O([TFSI]^−^), while ImCO2 and ImSO3 only caused a 6% reduction in those RDF peak heights. The more significant decrease in the RDF peak heights indicated the higher efficacy of ChoPO4 in diminishing Na^+^ coordination with [TFSI]^−^ and EO_10_.

Regarding the larger K^+^ ion, the ZW molecules had a weaker effect on K^+^ coordination preference for EO_10_ and [TFSI]^−^. Figure 3a,b shows that all ZW molecules caused a slight decrease in the peak heights of K^+^–O([TFSI]^−^) from 7 to 6 and exerted a minute effect on K^+^–O(EO_10_) RDFs. It was noted that while ChoPO4 had the strongest influence on the coordination preference of Na^+^/Li^+^ for [TFSI]^−^ and EO_10_, the three ZW compounds had a comparable effect on K^+^ coordination preference. Hence, compared to the effects of ImSO3 and ImCO2, ChoPO4 effect became weaker when increasing the cationic size.

The weakening of the ChoPO4 effect on K^+^ coordination preference might have been due to a decrease in K^+^ preference for coordinating with ChoPO4. Figure 3c indicates that Na^+^–O(ChoPO4) RDF peak was the highest. In contrast, K^+^–O(ChoPO4) RDF peak height was lower than that of K^+^–O(ImCO2) RDF peak. The larger size of K^+^ might hinder the coordination of the cation with all four O atoms in ChoPO4 molecules. Instead, the coordination with two O atoms from ZW molecules was preferred by K^+^. Appendix A shows that K^+^ preferred to coordinate with two O atoms having double bonds with the P atom in the PO_4_ group of ChoPO4. As shown in Appendix A, the RDF peak heights for K^+^ and O4/O5 were higher than those for K^+^ and O3/O6 in PO_4_ (O4/O5 are double-bonded to the P atom, and O3/O6 are single-bonded to the P atom). Moreover, as shown in Appendix A, the RDF peak heights of K^+^–O4/O5(ChoPO4) were higher than those of K^+^–O(ImCO2) and K^+^–O(ImSO3).

Figure 3c shows that increasing the cationic size led to a decrease in cation–ZW molecule coordination preference. The peak heights of the RDFs of K^+^–O(ZW) were lower than those of the RDFs of Na^+^–O(ZW), which were in turn lower than those of the RDFs of Li^+^–O(ZW). However, the coordination preference of cations for ChoPO4 showed a greater reduction than that for ImSO3 and ImCO2. For example, on the basis of our prior study on Li^+^ [24], the cation–O(ChoPO4) RDF peak height decreased by 50% when changing from Li^+^ to Na^+^, and by 61% when substituting Na^+^ with K^+^. The cation–O(ImSO3) RDF peak height decreased by 47% and 37% when changing from Li^+^ to Na^+^ and from Na^+^ to K^+^, respectively. Similarly, reductions of 38% and 32% were observed for the cation–O(ImCO2) RDF peak height when changing from Li^+^ to Na^+^ and from Na^+^ to K^+^, respectively. Such larger decrease in the coordination preference of cations for ChoPO4 compared to that for ImSO3 and ImCO2 was correlated with the weakening of the ChoPO4 effect on cation–O(EO_10_) and cation–O([TFSI]^−^) coordination preference.

As shown in Figure 3d, ChoPO4 and ImCO2 could induce the formation of indirect cationic pairs. The height of the RDF cation–cation peaks was increased with the addition of ChoPO4 and ImCO2 in both Na^+^ and K^+^ systems. The cation–cation RDF peak heights followed the order ChoPO4 > ImCO2 > ImSO3 for Na^+^, and ChoPO4 ≈ ImCO2 > ImSO3 for K^+^. This order of cation–cation RDF peak heights correlated with that of Na^+^–ZW and K^+^–ZW molecule RDF peak heights (Figure 3c and Appendix A), and a similar phenomenon was also evident in Li^+^ electrolytes [24]. This finding suggests that cations exhibit a strong tendency to coordinate with ZW molecules, which results in the aggregation of cations around ZW molecules. Consequently, ZW molecules function as “bridges,” facilitating indirect pairing among the cations themselves. The pairing of cations is illustrated in Appendix A.

### 3.2. Coordination Number

Table 2 and Table 3 show the calculated coordination numbers of cation–O(EO10) (N_cation-O(EO10)_), cation–O([TFSI]^−^) (N_cation-O([TFSI]−_), and cation–O(ZW) (N_cation-O(ZW)_) and their sums for Na^+^ and K^+^ in the eight electrolytes listed in Table 1. The total cation–O coordination number was higher for K^+^ (7.03–7.15) than for Na^+^ (6.52–6.70), which is consistent with the literature [44,45].

The highest coordination number was observed for cation–O(EO_10_). For example, as shown in Table 2, on average, a Na^+^ ion coordinated with 3.7 O atoms from EO_10_, 1.5 O atoms from [TFSI]^−^], and 1.5 O atoms from the ChoPO4 molecule. Similarly, the coordination numbers of K^+^–O exhibited the same trend as those determined for Na^+^ electrolytes (Table 3) as well as Li^+^ electrolytes [24]. While cations exhibited a stronger inclination to coordinate with the ZW molecules than with the other tested molecules (Figure 3), the coordination number of cation–O(ZW) was lower compared to that of cation–O(EO_10_), which was attributed to the low ZW molecule/cation molar ratio of 1:5. Our prior study [23] observed that when ImSO3 was added at an ImSO3/Li^+^ molar ratio of 1:1, the coordination number of Li^+^–O(ImSO3) was the highest in the ImSO3/LiTFSI/EO_5_ electrolyte at the Li^+^/O(EO_5_) molar ratio – r = 1:6.

Overall, adding ZW molecules to an electrolyte can reduce the coordination numbers of Na^+^–O and K^+^–O. Among the three examined ZW structures, ChoPO4 caused the highest decrease in cation–O([TFSI]^−^) and cation–O(EO_10_) coordination. For example, Table 2 shows that the N_Na+-O(EO10)_ decreased from 4.60 to 3.72 with ChoPO4, and to 4.03 and 4.11 with the addition of ImCO2 and ImSO3, respectively. Similarly, ChoPO4 addition also reduced the Na^+^–O([TFSI]^−^) coordination number by 28%, whereas ImCO2 reduced the coordination number by 20%, and ImSO3 by 17%. As shown in Table 3, the ZW molecules exerted the same effects in K^+^ electrolyte systems. N_K+-O(EO10)_ decreased from 4.83 to 4.27 with ChoPO4 and to 4.35 and 4.40 with the addition of ImCO2 and ImSO3, respectively. Likewise, the addition of ChoPO4 resulted in the highest reduction in the K^+^–O([TFSI]^−^) coordination number (ChoPO4: 24%; ImCO2: 20%; ImSO3: 17%). This trend indicated that ChoPO4 is better at reducing the coordination of cations with [TFSI]^−^ and EO_10_ compared to ImSO3 and ImCO2, which correlates with the higher preference of cations for forming coordination with ChoPO4 (as shown by the cation–O(ZW) RDF results—Figure 3).

As the cationic size increased, the effect of the ZW molecules on cation–EO_10_ weakened. Combining our previous study on Li^+^ [24] with Table 2 and Table 3, we observed that ChoPO4 decreased the N_cation-O(EO10)_ by 30% with Li^+^, by 19% with Na^+^, and only by 11% with K^+^. Similarly, ImSO3 and ImCO2 exhibited a comparable trend, reducing N_cation-O(EO10)_ by approximately 16% with Li^+^, 12% with Na^+^, and 10% with K^+^. The weakening of ZW molecules’ effect on cation–EO_10_ coordination number is consistent with the corresponding RDF results shown in Figure 3. The reduction in cation–ZW coordination preference led to a weakened effect of the ZW molecules on the cation-EO_10_ coordination number.

The effect of ChoPO4 on cation–O([TFSI]^−^) coordination weakened in the case of K^+^ compared to Na^+^, whereas ImCO2 and ImSO3 maintained their impact on cation–O([TFSI]^−^) coordination. Specifically, ChoPO4 decreased N_cation-O([TFSI]−)_ by 28% in the presence of Na^+^ and by 24% with K^+^. However, the reduction effect of ImCO2 and ImSO3 on N_cation-O([TFSI]−)_ remained consistent, amounting to 20% and 17%, respectively, for the Na^+^ and K^+^ systems. Our previous study [24] showed that ChoPO4, ImCO2, and ImSO3 reduced N_Li+-O([TFSI]_^−^_)_ by 29%, 17%, and 14%, respectively, which suggests ImCO2 and ImSO3 were even slightly more effective in reducing Na^+^/K^+^–[TFSI]^−^ than Li^+^–[TFSI]^−^ coordination.

The order of the cation–O(ZW) coordination numbers was as follows: ChoPO4 > ImCO2 > ImSO3. Table 2 shows that the Na^+^–O(ZW) coordination numbers were 1.47, 0.92, and 0.67 for ChoPO4, ImCO2, and ImSO3, respectively. Likewise, the K^+^–O(ChoPO4), K^+^–O(ImCO2), and K^+^–O(ImSO3) coordination numbers were 1.00, 0.96, and 0.77, respectively. This ranking of cation–O(ZW) coordination numbers aligns with their respective RDF peak heights. A higher cation–O(ChoPO4) coordination number accounted for a more substantial reduction in the coordination numbers of cation–O([TFSI]^−^) and cation–O(EO_10_) upon adding ChoPO4, compared to that observed in the presence of ImCO2 and ImSO3. The coordination number of cation–O(ImCO2) was higher than that of cation–O (ImSO3), which correlated with ImCO2 more effective dissociation effect on cation– O([TFSI]^−^) and cation–O(EO_10_) coordination.

With an increasing cation size, the cation–O(ChoPO4) coordination number decreased, while that of cation–O(ImCO2/ImSO3) increased. In combination with our study on Li^+^ electrolytes [24], Table 2 and Table 3 show that the cation–O(ChoPO4) coordination number was 1.69 for Li^+^,1.47 for Na^+^, 1.00 for K^+^. In contrast, the coordination number of cation–O(ImCO2) increased from 0.82 for Li^+^ to 0.92 for Na^+^, and further to 0.96 for K^+^. The same trend was observed for the cation–O(ImSO3) coordination number (0.60 for Li^+^, 0.67 for Na^+^, 0.77 for K^+^). The geometry of ChoPO4 might hinder its coordination with larger cations, resulting in a decrease in the cation–O(ChoPO4) coordination number with increasing cation size. ImSO3 and ImCO2 possess anionic groups at the end of their molecules, enabling an easier association with larger cations. It is noted that the cation–O(ImCO2) coordination number approached 1 as the cation size increased. This suggests that larger cations such as K^+^ might prefer to coordinate with one O atom from ImCO2 at the molar ratio K^+^/O(ImCO2) of 5:1.

### 3.3. Percentage of Li^+^ Leaving the Coordination with EO_10_ Chains

ChoPO4 demonstrated the highest efficacy in completely freeing Na^+^ and K^+^ from the coordination with EO_10_. In our prior study, we showcased the effectiveness of ZW molecules in releasing Li^+^ from the wrapping of EO chains [24]. In this study, we investigated this ZW molecules’ dissociation ability on Na^+^ and K^+^. As shown in Table 2 and Table 3, the ZW molecules decreased the average cation–O(EO_10_) coordination number, which indicated that the cations coordinated with PEO chains with fewer O atoms or were completely dissociated from the PEO chains. To further assess this dissociation effect of the ZW molecules, we computed the percentage of cations that were not in coordination with O(EO_10_) in the eight systems. Figure 4 shows that ChoPO4 addition led to a rise in the percentage of Na^+^ not coordinating with EO_10_, from 9% to 37%. ImCO2 and ImSO3 exhibited a lesser capacity to completely release Na^+^ from the coordination with EO_10_ chains. ImCO2 and ImSO3 dissociated around 27% and 21% of Na^+^ from the EO_10_ chains, respectively. As shown in Table 3 and Figure 4, these three ZW molecules showed the same trend for their ability to fully release K^+^ from EO_10_ chains. Such trend was correlated with the cation–O(ZW) and cation–O(EO_10_) coordination numbers. The more extensively the ZW molecules coordinated with cations, the better they dissociated cations from coordination with EO_10_.

The dissociation effect of ChoPO4 on cation–EO_10_ coordination decreased as the cation size increased, while that of ImSO3 and ImCO2 remained unvaried. Figure 4 shows that the percentage of cations not coordinating with EO_10_ decreased from 37% to 30% when ChoPO4 was added to the Na^+^ and K^+^ systems, respectively. In our previous work [24], we observed a higher percentage of Li^+^ (45%) not coordinating with EO_10_ in the presence of ChoPO4. The decreased effect of ChoPO4 correlated with the reduction in the cation–ChoPO4 coordination number. The larger cationic size favored the coordination with fewer O atoms from ChoPO4, thereby weakening the ability of ChoPO4 to replace EO_10_ in the cation coordination shell. However, ImSO3 and ImCO2 showed a minute change in their ability to release cations completely from coordination with EO_10_. ImSO3 and ImCO2 released 28% and 23% of Li^+^ from coordination with EO_10_, respectively, in our previous study [24].

ChoPO4 and ImCO2 had a weak exclusive solvation effect on Na^+^ and K^+^. Table 4 shows the percentages of Na^+^ and K^+^ coordinating exclusively with the ZW molecules. Around 2–3% of Na^+^ and K^+^ were coordinated exclusively with ChoPO4 and ImCO2. This exclusion effect of ChoPO4 was stronger in the case of Li^+^ [24]. Specifically, 9% of Li^+^ coordinated with ChoPO4 exclusively, at the same concentration and molar ratios as Na^+^ and K^+^ in their respective systems. The large ionic size hindered the coordination with multiple O atoms from ChoPO4, thus reducing the exclusive coordination with ChoPO4.

Na^+^ formed coordination with multiple O atoms from ChoPO4 to a greater extent than with O atoms from ImSO3 and ImCO2. Figure 5 illustrates the distribution of the cation–O(ZW) coordination numbers in the Na^+^ and K^+^ systems. The cation–O(ZW) coordination numbers were sorted by the number of O atoms in the ZW molecules with which the cation coordinated. Figure 5b shows that a higher percentage of Na^+^ was found to coordinate with more than one O atom from ChoPO4 than with more than one O atom from ImCO2 and ImSO3. Approximately 93% of Na^+^ coordinated with more than one O atom from ChoPO4, while around 83% of Na^+^ coordinated with more than one O atom from ImCO2, and that number was 45% for ImSO3 (7%, 17%, and 55% of Na^+^ coordinated with one O from ChoPO4, ImCO2, and ImSO3, respectively). The same trend was also observed with Li^+^ electrolytes [24].

The cation with larger size showed a reduction in the coordination with multiple O atoms from ChoPO4 and an increase in the coordination with multiple O atoms from ImSO3 and ImCO2. Figure 5b shows that the 80% of K^+^ coordinated with more than one O from ChoPO4, and this percentage was higher for Na^+^ (93%). Our previous study [24] observed that 100% of Li^+^ coordinated with more than one O from ChoPO4. In contrast, the percentage of cations coordinating with more than one O from ImSO3 increased from 30% for Li^+^ to 45% for Na^+^, and further to 55% for K^+^. Likewise, the percentage of cations coordinating with more than one O from ImCO2 was 70% and 83% for Li^+^ and Na^+^, respectively. However, the distribution of the cation–O(ImCO2) coordination numbers remained comparable between Na^+^ and K^+^. The trend of cations coordinating with multiple O(ZW) atoms was also reflected in the average cation–O(ZW) coordination numbers (Table 1 and Table 2). Appendix A show simulation snapshots as examples of the different morphology of the Na^+^–ZW molecule and K^+^–ZW molecule coordination systems.

### 3.4. Coordination Lifetime

Among the three examined ZW molecules, ChoPO4 caused the highest increase in the stability of Na^+^–O(EO_10_) and Na^+^–O([TFSI]^−^) coordination in Na^+^ electrolytes. As shown in Figure 6a, ChoPO4 increased τ_Na+–O(EO10)_ from 18 ns to around 37 ns, whereas τ_Na+–O(EO10)_ was around 23 ns when ImCO2 and ImSO3 were added. Likewise, Figure 6b shows that the presence of ChoPO4 resulted in an increase in τ_Na+–O(TFSI-)_ from 2 ns to around 5 ns, whereas τ_Na+–O([TFSI]−)_ remained constant around 2 ns with ImCO2 and ImSO3 addition. Such trend was also observed in Li^+^ electrolytes [24]. Since τ_Na+–O([TFSI]−)_ was more than one order of magnitude lower than τ_Na+–O(EO10)_ and τ_Na+–O(ZW)_, the increase in τ_Na+–O([TFSI]−)_ might not have significantly affected Na^+^ mobility compared to the enhanced stability of Na^+^–O(EO_10_) coordination.

Similarly, among the three ZW molecules, ChoPO4 enhanced the stability of K^+^–O(EO_10_) coordination the most. Figure 6a shows that τ_K+–O(EO10)_ increased from 7 ns to around 14 ns with ChoPO4 and to around 10 ns with ImCO2 and ImSO3 addition. However, the three ZW molecules had no effect on the stability of K^+^–O([TFSI]^−^) coordination. Figure 6b shows that τ_K+–O(TFSI-)_ remained around 2 ns in the presence of the ZW compounds. The increase in the stability of cation–O(EO_10_) coordination might have reduced cationic diffusion, as cations were affected by the slow dynamics of the EO chains.

Among the three ZW structures, cations established the most stable coordination with ChoPO4 and the least stable coordination with ImSO3. Figure 6c shows that τ_cation–O(ZW)_ was ranked in the order ChoPO4 > ImCO2 > ImSO3, reflecting the cation–ZW molecule coordination numbers. The lifetime of cation–O(ZW) coordination followed the same trend in K^+^ electrolytes. However, the lifetime of K^+^–O(ZW) coordination was lower than that of Na^+^–O(ZW) coordination. Our previous study [24] in Li^+^ showed higher cation–O(ZW) coordination lifetimes than in Na^+^. The decrease in cation–ZW molecule lifetime with increased cationic size correlated with the weakened preference of the cation to coordinate with ZW molecules. The reduced cation–O interaction lifetime for K^+^ compared to Na^+^ agrees with the literature [44] and can be attributed to the weaker cation–O interactions.

Except for ImSO3, cation–O(ZW) coordination was more stable than cation–O(EO_10_) and cation–O([TFSI]^−^) coordination. For instance, in Na^+^ electrolytes, τ_Na+–O(ChoPO4)_ was 202 ns, whereas τ_Na+–O(EO10)_ and τ_Na+–O(TFSI)_ were 37 ns and 5 ns, respectively. Similarly, τ_Na+–O(ImCO2)_ was 29 ns in Na^+^ electrolytes, exceeding τ_Na+–O(EO10)_ (24 ns) and τ_Na+–O(TFSI)_ (2 ns). However, cation–O(ImSO3) coordination was less stable than cation–O(EO_10_) coordination. For example, in Na^+^ electrolytes, τ_Na+–O(ImSO3)_ was 13 ns, which was lower than τ_Na+–O(EO10)_ of 37 ns. The same phenomenon was observed for K^+^ electrolyte. However, for all ZW molecules, Li^+^–O(ZW) showed higher stability compared to Li^+^–O([TFSI]^−^) and Li^+^–O(EO_10_) in Li^+^ electrolytes. The low Na^+^/K^+^–O(ImSO3) lifetime is related to the general decrease in cation–O(ZW) lifetime as the cationic size increases. Due to the larger molecular size of ZW structures compared to cations, the introduction of a stable cation–ZW molecule coordination can slow down cation diffusion. Thus, the reduction in cation–O(ZW) lifetime might lessen this slowing effect on the diffusion dynamics of large cations.

### 3.5. Diffusion Coefficients

Among the three examined ZW molecules, only ImCO2 enhanced the diffusion of Na^+^, while ChoPO4 and ImSO3 had a minimal effect on Na^+^ diffusion. Figure 7a shows that ImCO2 slightly increased the diffusion of Na^+^ by 20%, whereas ChoPO4 and ImSO3 slightly decreased Na^+^ diffusion by less than 10%. The greater diffusion enhancement by ImCO2 might be related to its better ability to release Na^+^ from EO_10_ compared to ImSO3 (Figure 4) and to weaken Na^+^–O(ImCO2) stability more than Na^+^–O(ChoPO4) stability (Figure 6c). Obviously, ImCO2 enhanced Na^+^ diffusion more effectively than ImSO3 due to the absence of a strong Na^+^ trap associated with Na^+^–ImCO2 coordination, as observed for ChoPO4.

Although all three ZW molecules reduced K^+^ diffusion, ChoPO4 had the greatest impact. As shown in Figure 7a, ChoPO4, ImSO3, and ImCO2 decreased K^+^ diffusion by 40%, 15%, and 15%, respectively. The highest reduction by ChoPO4 might be due to the higher K^+^–O(ChoPO4) coordination stability in comparison to the stability of the coordination between K^+^ and the other two ZW molecules (Figure 6). Previous research [23,24] highlighted the balance between two opposing effects of ZW molecules on Li^+^ diffusion: an enhancement due to the dissociation of Li^+^–O(EO) coordination and a deceleration due to the formation of stable Li^+^–O(ZW) coordination. We speculate that the decrease in the dissociation effect of ChoPO4 strengthened the diffusion-slowing effect, given the high cation–O(ChoPO4) stability. Consequently, ChoPO4 had the strongest impact on slowing K^+^ diffusion. Since ImCO2 and ImSO3 exhibited minimal changes in their cation–O(EO_10_) releasing ability with respect to Li^+^ and Na^+^, the weakened stability of the cation–ImCO2/ImSO3 coordination lessened their diffusion-slowing effect, leading to increased Na^+^ diffusion, in contrast to unvaried Li^+^, diffusion in the presence of ImCO2. Likewise, ImSO3 reducing effect on cationic diffusion was milder for Na^+^ (−10%) compared to Li^+^ (−25%). However, due to the weakening of the ZW molecules’ dissociation effect on the K^+^–O(EO_10_) coordination number (Table 2 and Table 3) and the enhancing of K^+^–O(EO_10_) stability (Figure 8a), both ImCO2 and ImSO3 decreased K^+^ diffusion.

As ZW molecules released cations from coordinated EO_10_ by forming the more preferred cation–ZW molecule coordination, we investigated the impact of this change in cationic coordination on the short-scale diffusion of cations. Figure 8 and Figure 9 illustrate the displacement of cations as a function of the cation–O coordination number (O from [TFSI]^−^, EO_10_, and ZW molecules) in Na^+^ and K^+^ electrolytes, respectively.

The coordination of cations with the ZW molecules slowed down the cation mobility. Figure 8 shows that the displacement distance of Na^+^ notably decreased upon its coordination with one O from the ZW molecules (Na^+^–O(ZW) coordination number equaling 1). Additionally, Na^+^ exhibited a decrease in short-scale motion with enhanced coordination with ZW molecules or [TFSI]^−^ compared to coordination with EO_10_. Similarly, the displacement of K^+^ also decreased with increasing K^+^–ZW coordination, as shown in Figure 9. Such phenomenon was likewise present in Li^+^ electrolytes [24]. This phenomenon also indicated that the introduction of ZW molecules hindered the vehicular movement of cations, as stable cation–ZW molecule interactions decelerated cation mobility. The aggregation of cations caused by ZW molecules might have also contributed to the decrease in cation transport. Similarly, Molinari et al. [46] reported that the clustering of Li^+^ and [TFSI]^−^ was associated with shorter displacements of Li^+^ in a LiTFSI/EO_100_ system at 4 M Li^+^ concentration. Meanwhile, ZW molecules might be beneficial for the cation hopping mechanism, which necessitates the transition between coordination sites and can be induced by the dissociation effect of ZW molecules.

### 3.6. Radius of Gyration of EO_10_

ChoPO4 and ImCO2 molecules can enhance the coiling of EO_10_ chains. Here, we analyzed the EO_10_ conformation using the radius of gyration (*R_g_*), a measure of the average coil size of EO_10_ chains. Figure 10 shows that the *R_g_* of the EO_10_ chains decreased slightly due to the addition of ChoPO4 and ImCO2 structures in both Na^+^ and K^+^ cationic systems. The lower *R_g_* aligned with the higher percentage of Na^+^ ions dissociated from the coordination with the EO_10_ chains (Figure 4). The same phenomenon was also present in K^+^ systems. However, ZW molecules had an opposite effect in Li^+^ systems. Our previous study [24] showed that the addition of ZW molecules relaxed the EO chain coiling in Li^+^ electrolytes. The ZW molecules reduced the wrapping of the EO_10_ chains around Li^+^, resulting in an increase in the *Rg* of the EO_10_ chains. However, given the larger ionic size of Na^+^ and K^+^, there might be a competition between the EO_10_ chains for the coordination of Na^+^/K^+^ cations. Each Na^+^/K^+^ cation can coordinate with approximately seven O atoms. On average, our systems had six O(EO_10_) atoms available for each cation. Thus, there were some EO_10_ chains that did not completely coil around the cations. By releasing certain cations from the wrapping of EO_10_, the remaining EO_10_ chains would have a greater possibility to coordinate with the cations, which would lead to more tightly coiled chains. Hence, ZW molecules could potentially influence the cationic transport dynamics in PEO electrolytes by altering the conformation of the PEO chains. The *R_g_* distribution for the EO_10_ chains is shown in Appendix A.

### 3.7. Cationic Charge Density

Table 5 lists the ionic radius and surface charge density of cations (Li^+^, Na^+^, K^+^) and Pauling electronegativity of Li, Na, and K elements. As the cation size increases from Li^+^ to K^+^, the surface charge density decreases, leading to a weaker cation–O(ZW) coordination. Likewise, the decrease in Pauling electronegativity agrees with the reduced ability of cations to interact with O atoms from anions, ZW molecules, and (EO)_10_ molecules. The weakened cation–O interactions observed for Li^+^ to K^+^ is consistent with the literature. Okoshi et al. [47] conducted density functional theory (DFT) calculations, revealing a diminishing interaction energy (desolvation energy) with 27 organic solvents as the cationic size increased, following the order Li^+^ > Na^+^ > K^+^, aligning with the Lewis acidity of the respective cations. Similarly, Pham et al. conducted simulation studies on Li^+^, Na^+^, and K^+^ solvation in ethylene carbonate, establishing the order of solvation energies Li^+^ > Na^+^ > K^+^ (5.85, 4.72, and 4.12 eV, respectively) [45]. Additionally, Wróbel et al. reported analogous results, highlighting lower binding energies for Na^+^ to the carbonate molecule compared to Li^+^ [43]. This weaker interaction of Na^+^ with solvents compared to Li^+^ is consistent with a comprehensive theorical–experimental study by Creasce et al. [48].

In our study, weaker interactions were reflected in the decrease in cation–O(ZW) RDF peak heights and stability (Figure 3 and Figure 6). Likewise, the dissociation effect of the ZW molecules weakened in relation to the cation–O(EO)_10_ coordination (Table 2 and Table 3). ChoPO4 particularly showed the lowest ability to fully release cations from coordination with EO_10_ due to Na^+^/K^+^ preference for single O atom coordination over coordination of multiple O atoms from ZW molecules (Figure 5). Meanwhile, the weakening of cation–O(ZW) coordination stability lessened the decelerating effect of the ZW molecules on cationic diffusion. As demonstrated in Figure 7, ImCO2 enhanced Na^+^ diffusion by 20%, and ImSO3 had a milder impact on Na^+^/K^+^ diffusion compared to Li^+^ diffusion.

### 3.8. Effect of the Simulation Temperature

To explore the effect of the simulation temperature on cation diffusion, we performed similar MD simulation analyses at T = 600 K. We demonstrated that an increase in temperature from 353 K to 600 K enhanced the mobility of ions and molecules while preserving the solvation structures of Na^+^ and K^+^. This temperature elevation was essential for the investigation of the dynamics of ions and molecules within a practical simulation timeframe. Appendix A show that ions and molecules exhibited slower dynamics in systems at 353 K compared to those at 600 K (Appendix A). For instance, the C(t) curve of the Na^+^–O(EO_10_) coordination decayed to 80–90% after 500 ns in a system at 353 K (Appendix A), whereas it decreased to 0% in less than 300 ns at 600 K (Appendix A). Furthermore, Appendix A indicates that the diffusion coefficients of Na^+^/K^+^ at 353 K ranged from 10^−5^ to 10^−4^ nm^2^/ns, which are two orders of magnitude lower than those observed at 600 K (Appendix A). Nevertheless, the ionic solvation of cations at 353 K and 600 K remained comparable. Appendix A show that ChoPO4 exerted the most pronounced influence in reducing the coordination numbers of cation–O([TFSI]^−^) and cation–O(EO_10_) for systems at 353 K. Moreover, Appendix A indicates that the ZW molecules fully released cations from coordination with EO_10_ in the order ChoPO4 > ImCO2 > ImSO3 at 353 K. Likewise, at 353 K, the ranking of the RDF peak heights and coordination numbers of cation–O(ZW) followed the order ChoPO4 > ImCO2 > ImSO3. Additionally, the Na^+^–Na^+^ pairing preference at 353 K was enhanced with the addition of the ZW molecules in the order ChoPO4 > ImCO2 > ImSO3 (Appendix A). However, differences in the effect of the ZW molecules were observed between systems at 353 K and 600 K. For instance, the K^+^–K^+^ RDF heights in ChoPO4 presence were comparable to those observed with ImCO2 addition at 353 K (Appendix A). Additionally, the impact of the ZW molecules on the dissociation of K^+^–EO_10_ was more significant than that on the dissociation of Na^+^–EO_10_ (Appendix A). The elevated temperature improved the ion dynamics, consequently rendering the data of ionic solvation more reliable.

## 4. Conclusions

This work investigated the effect of three ZW molecules (ChoPO4, ImCO2, and ImSO3) on Na^+^ and K^+^ coordination in NaTFSI/EO_10_ and KTFSI/EO_10_ systems using MD simulations. We analyzed the structural and dynamic properties of ionic solvation and diffusion in eight electrolytes composed of NaTFSI/EO_10_ and KTFSI/EO_10_ without ZW molecule and with ZW molecule addition (ZW/Na^+^ (or /K^+^) = 1:5) at the Na^+^ (or K^+^)/ O(EO_10_) molar ratio – r = 1:6.

The simulation results unveiled both common and distinctive effects of the three ZW molecules on the solvation of Na^+^ and K^+^ in NaTFSI/EO_10_ and KTFSI/EO_10_ systems. ChoPO4, ImSO3, and ImCO2 exhibited the ability to detach Na^+^ and K^+^ from the EO_10_ chains through the formation of cation–ZW molecule coordination, with the order of effectiveness being ChoPO4 > ImCO2 > ImSO3. The specific impact of these ZW molecules is attributed to their anionic groups. Among the three structures, ChoPO4, with the PO_4_ group possessing the highest electronegativity, formed the most stable cation–ZW molecule coordination and exerted the strongest dissociation effect on the cation–EO_10_ system. ImCO2, with the CO_2_ group characterized by the highest charge density, ranked second in terms of cation–ZW molecule coordination stability and its effect on cation–EO_10_ dissociation. As the cationic size progressed from Li^+^ to Na^+^ to K^+^, the dissociating effect of the ZW molecules on cation–EO_10_ coordination weakened due to a diminished preference of cations for coordinating with the ZW molecules. Notably, only ChoPO4 experienced a drop in its ability to fully release K^+^ from EO_10_ wrapping, as compared to Na^+^ and Li^+^. The larger K^+^ cations appeared to prefer to coordinate with two atoms from ChoPO4 as opposed to all four. Among the three examined ZW molecules, ChoPO4 was also the sole structure to demonstrate a reduction in cation–O(ChoPO4) coordination number as the cationic size increased from Li^+^ to Na^+^ and to K^+^. The overall impact of the ZW molecules on cationic diffusion appeared to be a balance between the ZW molecules’ accelerating effect due to the dissociation of the cation–EO_10_ system and the ZW molecules’ decelerating effect due to the stable cation–ZW molecule coordination. Consequently, the diminishing dissociation effect of ChoPO4 resulted in a mild decelerating influence on Na^+^ diffusion and a most pronounced slowing effect on K^+^ diffusion. Conversely, ImCO2 and ImSO3 demonstrated minute changes in their ability to fully release cations from EO_10_ in Li^+^, Na^+^, and K^+^ systems. Moreover, the weakened stability of cation–ImCO2/ImSO3 coordination reduced the ZW molecules’ decelerating effect, resulting in an increase in Na^+^ diffusion compared to unchanged Li^+^ diffusion in the presence of ImCO2. Similarly, the ImSO3 decelerating effect on cationic diffusion was less pronounced for Na^+^ compared to Li^+^ due to the reduced cation–ImSO3 system stability. Nonetheless, due to the further weakening of the ZW molecules’ dissociation effects on the K^+^–O(EO_10_) coordination number and the enhancement of K^+^–O(EO_10_) stability, both ImCO2 and ImSO3 led to a similar reduction in K^+^ diffusion. Our simulations suggest that the impact of ZW molecules on cationic transport depends on the intricate interplay between various cationic coordination properties, such as cation–EO_10_ dissociation effect, cation–ZW molecule stability, and cation–EO_10_ stability. The understanding of these interactions is instrumental in designing ZW molecular structures as additives to regulate Na^+^ and K^+^ ionic solvation in polymer systems. ZW molecules may exhibit enhanced performance in a polymer system with an optimized balance of dissociation effect and decelerating effect, as observed with the addition of ImCO2 to Na^+^ electrolytes.

## Figures and Tables

**Figure 1 nanomaterials-14-00219-f001:**
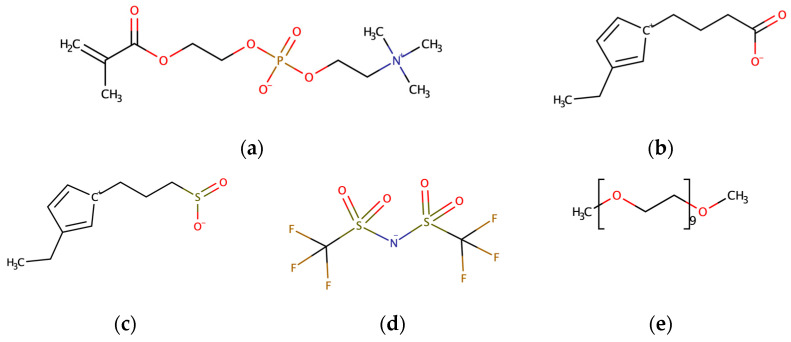
Molecular structures of (**a**) ChoPO4, (**b**) ImSO3, (**c**) ImCO2, (**d**) [TFSI]^−^, and (**e**) EO_10_.

**Figure 2 nanomaterials-14-00219-f002:**
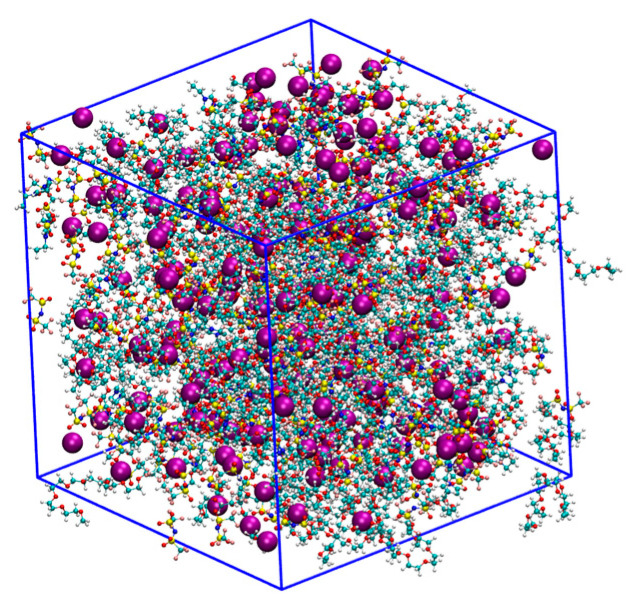
The configuration of the simulation box containing 200 Na^+^ (or K^+^) ions, 200 [TFSI]^−^ ions, 40 ImSO3 molecules, and 120 EO_10_ molecules. Na^+^/K^+^ ions are represented using the VDW model, and the others are represented by the CPK model. Atom color code: Na+/K+ ions (purple), hydrogen (silver), carbon (cyan), nitrogen (blue), oxygen (red), sulfur (yellow). This figure was generated using VMD [28].

**Figure 3 nanomaterials-14-00219-f003:**
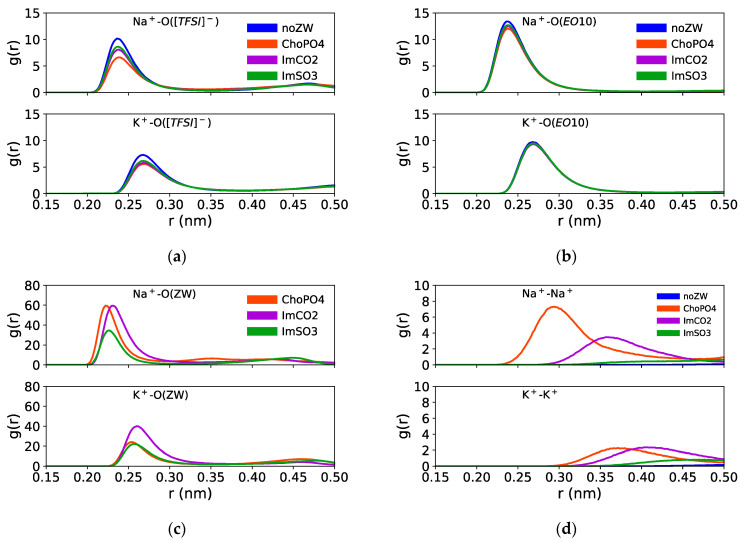
RDFs of (**a**) Na^+^/K^+^–O([TFSI]^−^), (**b**) Na^+^/K^+^–O(EO_10_), (**c**) Na^+^/K^+^–O(ZW), and (**d**) Na^+^–Na^+^ and K^+^–K^+^ in Na^+^/K^+^ systems (g(r) were normalized to 1 at long distance).

**Figure 4 nanomaterials-14-00219-f004:**
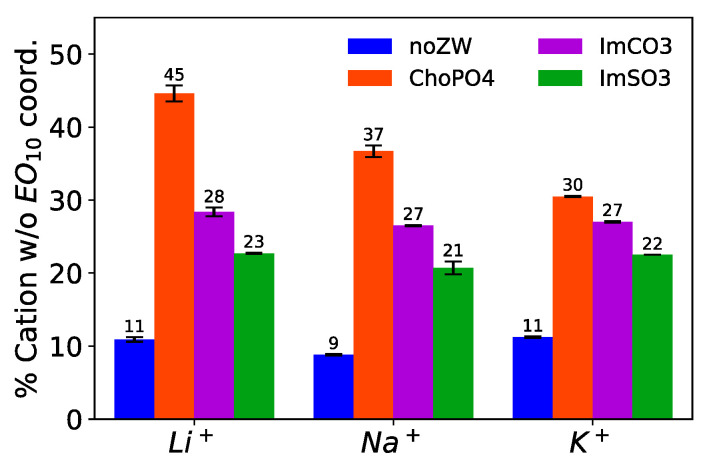
Percentage of cations not coordinating with EO_10_. Note: Data for the Li^+^ systems are adapted from our previous study [24].

**Figure 5 nanomaterials-14-00219-f005:**
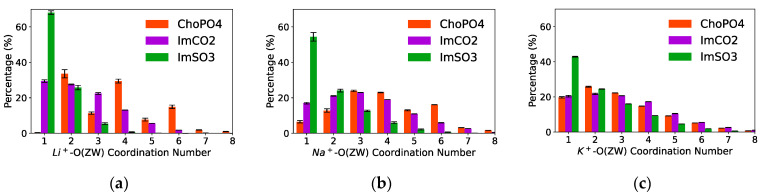
Distribution of cation^+^–O(ZW) coordination numbers in (**a**) Li^+^, (**b**) Na^+^, and (**c**) K^+^ systems. Note: Data for the Li^+^ systems are adapted from our previous study [24].

**Figure 6 nanomaterials-14-00219-f006:**
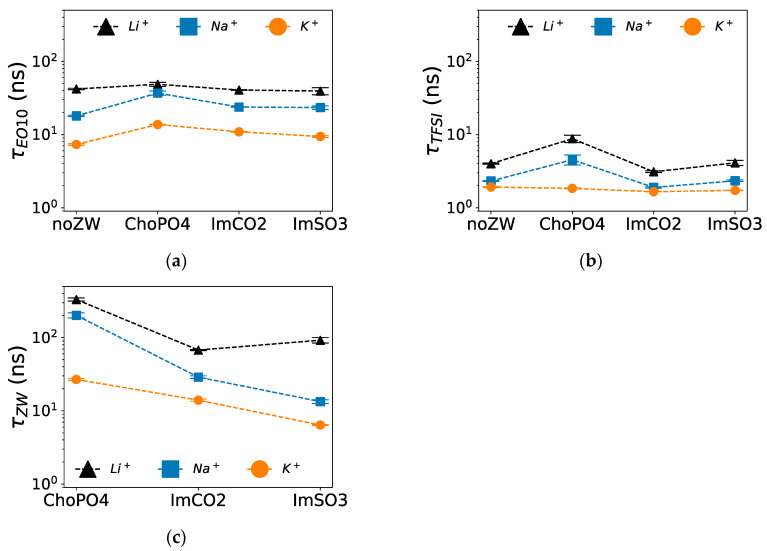
Lifetime of cation–O coordination in Li^+^, Na^+^, and K^+^ systems. (**a**) Cation^+^–O(EO_10_), (**b**) cation^+^–O([TFSI]^−^), and (**c**) cation^+^–O(ZW) coordination. Note: Data for the Li^+^ systems are adapted from our previous study [24].

**Figure 7 nanomaterials-14-00219-f007:**
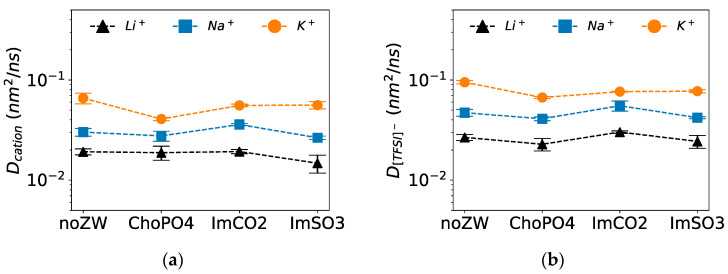
Diffusion coefficients of (**a**) cations and (**b**) [TFSI]^−^ at 600 K. Note: Data for the Li^+^ systems are adapted from our previous study [24].

**Figure 8 nanomaterials-14-00219-f008:**
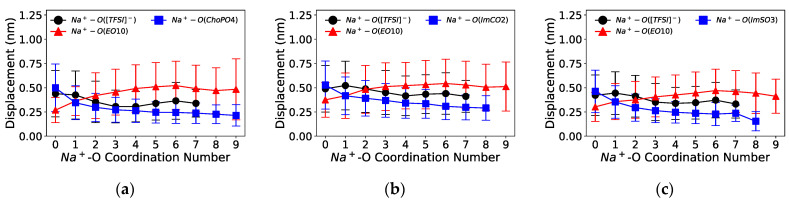
Distance travelled by Na^+^ ions as a function of the O coordination number over 1 ns for systems (**a**) with ChoPO4, (**b**) ImCO2, and (**c**) ImSO3.

**Figure 9 nanomaterials-14-00219-f009:**
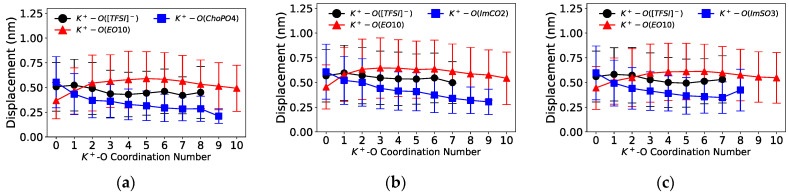
Distance travelled by K^+^ ions as a function of the O coordination number over 1 ns for systems (**a**) with ChoPO4, (**b**) ImCO2, and (**c**) ImSO3.

**Figure 10 nanomaterials-14-00219-f010:**
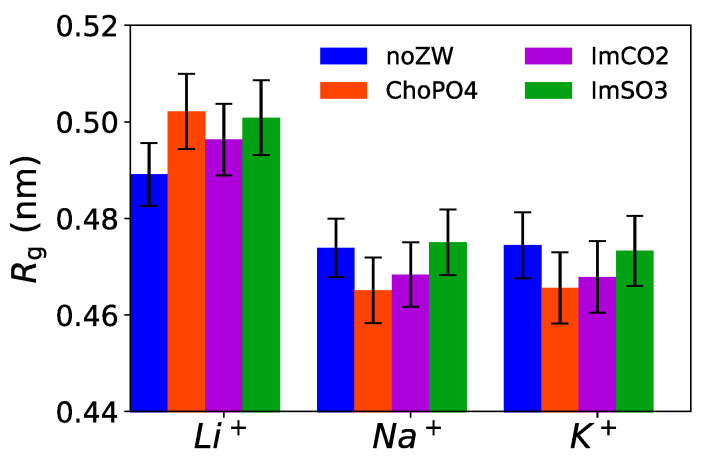
Radius of gyration (Rg) of the EO10 chains in the Li^+^, Na^+^, and K^+^ systems. Note: Data for the Li^+^ systems are adapted from our previous study [24].

**Table 1 nanomaterials-14-00219-t001:** Details of the simulation systems.

#	Na^+^/K^+^	[TFSI]^−^	EO_10_	ZW Molecule	Na^+^/ or K^+^/ O(EO_10_)	Mol. Conc. (mol/L)	Total Number of Atoms
1	200	200	120		1:6	2.3	11,840
2	200	200	120	40 ChoPO4	1:6	2.3	13,480
3	200	200	120	40 ImCO2	1:6	2.4	12,920
4	200	200	120	40 ImSO3	1:6	2.4	12,960

**Table 2 nanomaterials-14-00219-t002:** Coordination numbers of Na^+^–O.

	No ZW Molecule	ChoPO4	ImCO2	ImSO3
Total	6.70 ± 0.01	6.70 ± 0.01	6.63 ± 0.01	6.52 ± 0.01
Na^+^–O(ZW)	N/A	1.47 ± 0.01	0.92 ± 0.01	0.67 ± 0.02
Na^+^–O([TFSI]^−^)	2.11 ± 0.01	1.51 ± 0.01	1.69 ± 0.01	1.75 ± 0.01
Na^+^–O(EO_10_)	4.60 ± 0.01	3.72 ± 0.03	4.03 ± 0.01	4.11 ± 0.01

**Table 3 nanomaterials-14-00219-t003:** Coordination numbers of K^+^–O.

	No ZW Molecule	ChoPO4	ImCO2	ImSO3
Total	7.15 ± 0.01	7.03 ± 0.01	7.16 ± 0.01	7.09 ± 0.01
K^+^–O(ZW)	N/A	1.00 ± 0.01	0.96 ± 0.01	0.77 ± 0.01
K^+^–O([TFSI]^−^)	2.33 ± 0.01	1.77 ± 0.01	1.86 ± 0.01	1.92 ± 0.01
K^+^–O(EO_10_)	4.83 ± 0.01	4.27 ± 0.01	4.35 ± 0.01	4.40 ± 0.01

**Table 4 nanomaterials-14-00219-t004:** Percentage of cations coordinating exclusively with the ZW molecules.

ZW Structures	Systems		
	Li^+^	Na^+^	K^+^
ChoPO4	9 ± 1%	2.9 ± 0.1%	2.2 ± 0.1%
ImSO3	0	0.6 ± 0.2%	0.6 ± 0.1%
ImCO2	2 ± 1%	2.7 ± 0.1%	1.8 ± 0.1%

Note: Data for the Li^+^ systems are adapted from our previous study [24].

**Table 5 nanomaterials-14-00219-t005:** Charge density of cations.

	Li^+^	Na^+^	K^+^
Ionic radius (Å)	0.76	1.02	1.38
$\mathrm{Surface}\;\mathrm{charge}\;\mathrm{density}\;(e/{Å}^{2}$)	0.14	0.08	0.04
Pauling electronegativity	0.98	0.93	0.82

## Data Availability

Data are contained within the article.

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
