# Peer review of "Effect of Zwitterionic Additives on Solvation and Transport of Sodium and Potassium Cations in (Ethylene Oxide)10: A Molecular Dynamics Simulation Study"

_nanomaterials, 2024, doi:10.3390/nano14020219_

Round 1
Reviewer 1 Report
Comments and Suggestions for Authors
The manuscript, entitled by “Effect of Zwitterionic Additives on Solvation and Transport of Sodium and Potassium Cations in (Ethylene Oxide)10: a Molecular Dynamics Simulation Study” (nanomaterials-2742706), investigated the dissociation effects of three ZW molecules on Na+ and K+ in ethylene oxide through molecular simulations. The author provided lots of results, the presentation and discussion were similar with the previously publication (Ref. 24 distinct effects of zwitterionic molecules on ionic solvation in (ethylene oxide)10: a molecular dynamics simulation study). So, the deep discussion of difference between the Li+ and Na+/K+ have been expected in current work. The manuscript could be recommended for publication after addressing following issues:
1. More discussions should be added in results section. Moreover, only two publications, both from authors’ group have been cited in current version. More experimental and theoretical works should be added in this section to validate the simulated results.
2. The difference behaviors between Li+ and Na+/K+ have been simply attributed to the ionic size. More factors should be considered, including chemical activity, electronegativity, etc.
3. What boundary conditions did the authors adopted in the simulation? As seen in Fig.2, there is vacuum region in the simulation box. How to deal with the periodic interaction?
4. For improving the understandability and readability, the analysis methods should be briefly introduced in Methodology, including Radial distribution functions, etc. The equation for C(t) curve (Eq.(2)) should also be moved to Sec. 2.
Reviewer 2 Report
Comments and Suggestions for Authors
This paper by Shao and coworkers investigates mixtures of short-chain ethylene oxide, sodium/potassium TFSI salts, and zwitterions through molecular dynamics simulations. This work is situated within an impactful area of polymer science, its focus on zwitterions is particularly well timed due to recent advances in this area. Herein these workers focus on the coordination and dynamics of these mixtures. The structure-property trends that the researchers find are somewhat predictable based on other work in this area, yet still would be valuable to workers in this field. This is a high-quality work, however significant modifications could be made to improve the publication – some suggestions are included below.
· Page 1 line 10: “Solid polymer electrolytes (SPEs) are essential 10 for safer and more efficient Na+ and K+ batteries because they often exhibit low ionic conductivity 11 at room temperature” – I do not think this statement is what was intended here, low ionic conductivity is a disadvantage of SPEs, not an advantage.
· Page 1 line 13-15: “we investigated the 13 dissociation effects of three ZW molecules (MPC: 2-methacryloyloxyethyl phosphorylcholine, SB: 14 sulfobetaine ethylimidazole, CB: carboxybetaine ethylimidazole) on Na+ and K+ in ethylene oxide” – this is unclear, what are the ‘dissociation effects’? Please clarify this in the abstract.
· I would strongly suggest that the zwitterions are given names and that the structures are presented at least one time in a form that is easily readable by an organic chemist. Figure 1 is the only place that these structures are presented and in this figure it is challenging to determine what the actual chemistries are. I would suggest that you replace these ball-and-stick representations with more typical chemistry drawings. Further, I would name the molecules according to their relevant chemistries, rather than complex chemical abbreviations, this will allow readers to understand diagrams significantly better. E.g. instead of calling [sulfobetaine ethylimidazole] SB, I would call it Eth-Im-C3-SO3, or something similar that gives the reader information about the molecule, rather than just being an acronym to reference back to the abstract.
· Figure 2 does not seem like a worthwhile figure to include. There is nothing the audience can learn from looking at this figure and you perform no analysis and make no points based on this figure.
· It would be helpful to present the ionic concentrations in terms of ‘r’-ratios as this is standard in the polymer electrolyte field, where r=moles of salt/moles of monomer
· Figures 3D and 4D have different x-axis distances than A, B and C. I find it is a bit confusing since nothing of much interest happens beyond r=0.5 nm. It would be prudent to make all x-value ranges the same so that these plots are easier to compare. It may also prudent to stack all plots vertically rather than the 4-panel style chosen so that it would be easier to compare the coordination distances of these different atomic pairs.
· The authors make some comparisons between figures 3D and 4D in a few places. I think it would be a lot easier to understand the effects of ion size if some figures were included that have both Na and K on the same RDF plot. Perhaps it would be possible to plot the RDF functions as two stacked plots?
· Page 7 line 211: What are the sizes of these cation/ZI clusters? This could be essential to understand. Are these clusters just ion pairs as you seem to indicate or are they actually dense ionic clusters of many cations? I would more discussion of the implication of figures 3d and 4D generally… what is the implication of these cationic pairings?
· Page 7, final paragraph: The coordination number has some serious implications on transport, I would make these connections to ion transport more clear in this part.
· Figure 5 – The y label needs to have a better title so that it can be read at a glance
· Figure 5- Unclear how a molecule that is ‘completely free of EO’ is defined, is it a molecule that doesn’t have EO in the first coordination shell? I think it would be good to make this quantity clearer.
· Figures 7 and 8 – these snapshots are not discussed in the text at all. These figures should be omitted or some significance need to be attributed to them in the main text of the article.
· Figure 10 – it would be good to show the displacement verus time portfolios in the SI so that it can be verified that the diffusion coefficient comes from a diffusive behavior regime. It is fairly common for there to be subdiffusive and diffusive regimes, so this is quite an important validation.
· Figure 13 – I am not sure that the analysis of this figure stands up to interpretation. It seems that there can be no statistically significant zwitterion effect on Rg so I am surprised to read the interpretation.
Comments on the Quality of English LanguageEnglish is of reasonably good quality.
Round 2
Reviewer 1 Report
Comments and Suggestions for Authors
The authors have addressed all of my original concerns with the manuscript and have added additional references, which has improved the overall quality of the work. The article is recommened for publication.